# Liquid Biopsy in Peritoneal Carcinomatosis from Colorectal Cancer: Current Evidence and Future Perspectives

**DOI:** 10.3390/cancers17091461

**Published:** 2025-04-26

**Authors:** Valentino Martelli, Joana Vidal, Sílvia Salvans, Concepción Fernández, Jordi Badia-Ramentol, Jenniffer Linares, Marta Jiménez, Annarita Sibilio, Joan Gibert, Marina Pérez, Beatriz Bellosillo, Alexandre Calon, Filippo Pietrantonio, Mar Iglesias, Marta Pascual, Clara Montagut

**Affiliations:** 1Medical Oncology Department, Hospital del Mar, Hospital del Mar Research Institute, Universitat Pompeu Fabra, Centro de Investigación Biomédica en Red Cancer (CIBERONC), 08003 Barcelona, Spain; martellivalentino91@gmail.com (V.M.); jlinares@researchmar.net (J.L.); asibilio@researchmar.net (A.S.);; 2Department of Internal Medicine and Medical Specialties (DiMI), School of Medicine, University of Genova, 16133 Genova, Italy; 3Section of Colon and Rectal Surgery, Department of Surgery, Hospital del Mar, 08003 Barcelona, Spain; ssalvans@parcdesalutmar.cat (S.S.); mpascual@hmar.cat (M.P.); 4Pathology Department, Hospital del Mar, Hospital del Mar Research Institute, Universitat Pompeu Fabra, Centro de Investigación Biomédica en Red Cancer (CIBERONC), 08003 Barcelona, Spain; mconcepcionfernandezrodriguez@psmar.cat (C.F.); jgibert@psmar.cat (J.G.); bbellosillo@hmar.cat (B.B.); miglesiasc@hmar.cat (M.I.); 5Cancer Research Program, Hospital del Mar Research Institute, 08003 Barcelona, Spainmjimeneztoscano@hmar.cat (M.J.); acalon@researchmar.net (A.C.); 6Fondazione Istituto di Ricovero e Cura a Carattere Scientifico (IRCCS) Istituto Nazionale dei Tumori, 20133 Milan, Italy; filippo.pietrantonio@istitutotumori.mi.it

**Keywords:** circulating tumor DNA, metastatic colorectal cancer, peritoneal tumor DNA, peritoneal carcinomatosis, precision medicine

## Abstract

Peritoneal carcinomatosis from colorectal cancer (PC-CRC) is a challenging condition associated with poor outcomes, even with aggressive treatments like cytoreductive surgery (CRS) and hyperthermic intraperitoneal chemotherapy (HIPEC). One of the main challenges is the lack of reliable biomarkers to guide personalized treatment decisions. Circulating tumor DNA (ctDNA) in plasma has recently emerged as a promising tool in colorectal cancer (CRC) management. It detects residual disease after surgery in early-stage CRC and tracks mutations, treatment response, and resistance in metastatic CRC (mCRC). For PC-CRC, ctDNA, along with peritoneal tumor DNA (ptDNA) detected in peritoneal fluid, could offer significant insights for guiding treatment. These biomarkers have potential to improve disease monitoring and outcomes in this challenging context. This review examines the strengths, limitations, and future applications of liquid biopsy for PC-CRC care.

## 1. Introduction

Colorectal cancer (CRC) is the third most frequent cancer worldwide and the second leading cause of cancer-related death [1]. Up to 25% of patients are diagnosed after the disease has already metastasised, when palliative chemotherapy is the standard treatment [2,3]. Among metastatic sites, the peritoneum (i.e., the serous membrane lining the abdominal cavity and covering the abdominal organs) is affected in a small percentage of cases [4]. Epidemiological data on peritoneal carcinomatosis (PC) of colorectal (PC-CRC) origin have only recently been obtained through large population-based studies [5], reporting a prevalence of synchronous carcinomatosis of up to 4%, very close to the prevalence of metachronous disease [6]. However, these rates are likely underestimated, particularly for metachronous carcinomatosis, which can reach up to 40% [7].

The dissemination of tumor cells within the peritoneal cavity can occur through different mechanisms, including intraperitoneal spread by contiguity, hematogenous spread, lymphatic dissemination, and iatrogenic spread during surgical procedures [8].

This process is facilitated by the elevated interstitial fluid pressure typical of solid tumors, which is a result of rapid cellular proliferation, defective lymphatic drainage, and fibrosis in the interstitial matrix [9]. Additionally, the omentum is a preferential site for peritoneal tumor growth. Although the reasons for this “tropism” remain incompletely understood, it is thought that tumor growth is stimulated by fatty acids stored in omental adipocytes and by the pro-angiogenic environment of the omental “milky spots”, which consist of immune cell aggregates and a dense capillary network [10].

Challenges in identifying PC-CRC are mainly due to the low sensitivity and specificity of current standard imaging techniques in detecting tumor deposits in the serosa, since gross tumor lesions are rare findings. Thus, in most of the cases, metastases are represented by small nodules (usually smaller than 1 cm) that follow the anatomic outline of the surrounding organs, which are difficult to visualize by CT or PET-CT scan [11]. Furthermore, the clinical presentation of PC-CRC can be misleading: nausea, vomiting, abdominal pain, and bloating are the common signs and symptoms of this potentially life-threatening scenario, which can lead to intestinal obstruction [12,13]. Finally, the limitation to accurately measure peritoneal lesions and their response to treatment poses significant challenges for the inclusion of PC-CRC patients in randomized clinical trials [14].

In this context, efforts have been made in the past decades to improve the diagnosis and treatment of PC-CRC, with encouraging advances both in imaging techniques, biomarkers, and novel therapeutic strategies.

Cytoreductive surgery (CRS) is the current cornerstone of curative treatment for isolated PC-CRC, aiming to eradicate all macroscopic disease. After a midline laparotomy, a thorough exploration of the abdominal cavity is conducted to determine the extent of the disease using the peritoneal cancer index (PCI), the main method for mapping the distribution of metastases within the peritoneal cavity [15]. Subsequently, peritonectomy and visceral resections are performed to achieve complete resection of the disease [16]. Once these procedures are completed, the completeness of cytoreduction (CC) score is assessed [17]: CC-0 (no residual nodules), CC-1 (<2.5 mm), CC-2 (<25 mm), and CC-3 (>25 mm). Complete cytoreduction, defined as CC-0 or CC-1, has been shown to be an independent prognostic factor [18]. However, even in these cases, microscopic disease may persist, necessitating additional treatments to minimize the risk of relapse.

To address the microscopic residual disease, CRS is commonly followed by HIPEC, aimed at eradicating any remaining tumor cells [19]. Originally developed as a postoperative treatment, HIPEC is now administered intraoperatively, following CRS [20]. Two main delivery methods are currently used: the closed-abdomen technique and the open-abdomen or “coliseum technique” [21]. In both approaches, heated chemotherapy is circulated within the abdominal cavity for a set period, typically ranging from 30 to 90 min, allowing direct contact with tumor cells while minimizing systemic exposure [22]. The choice of cytotoxic agents and their dosing varies globally, but mitomycin C and oxaliplatin [23] are the most commonly employed [24]. Despite its promise as a localized treatment, HIPEC can be associated with potential serious side effects, including abdominal bleeding, hematologic toxicities, and acute kidney injury [25,26]. The most recent evidence on CRS + HIPEC comes from the French PRODIGE 7/ACCORD-15 study, which randomized 267 PC-CRC patients who had previously undergone systemic chemotherapy to receive either CRS with or without oxaliplatin-based HIPEC. Unfortunately, the study did not meet its primary endpoint, as no statistically significant differences in overall survival (OS) were observed between the two groups (hazard ratio, HR 1; *p* = 0.995) [27]. However, the statistical methodology and the chemotherapy agent used may have negatively influenced the results [28]. Other studies, using different treatment regimens, have demonstrated the efficacy of CRS + HIPEC, with some reporting a 5-year recurrence-free survival rate of 16% and a median overall survival (mOS) of up to 30 months, although outcomes vary considerably in the literature [29].

Identifying patients with the highest likelihood of achieving long-term survival is crucial to avoid unnecessary toxicities and potentially ineffective invasive treatments. To this end, the discovery of novel prognostic and predictive biomarkers is essential [30]. A growing body of research supports the use of circulating tumor DNA (ctDNA; so-called liquid biopsy) in oncology, with potential applications in early diagnosis, residual disease detection, relapse prediction, treatment response monitoring, and tracking tumor clonal evolution [31]. However, the clinical use of ctDNA in PC-CRC remains a challenge, mainly due to small evidence on the use of ctDNA in PC-CRC and low ctDNA detection rates in this clinical scenario [32].

In this review, we will provide an overview of the main features and available technologies for ctDNA analysis. In particular, we will focus on the role of ctDNA in PC-CRC, its utility in detecting early peritoneal relapse, and its potential predictive and prognostic value in advanced disease. Moreover, in this review we will assess for the first time the possible applications and future clinical implications of peritoneal fluid as an alternative source of ctDNA genotyping (peritoneal tumor DNA; ptDNA), hypothesizing different clinical scenarios obtained through the integration of ctDNA and ptDNA.

## 2. Materials and Methods

This narrative review was conducted to synthesize the current knowledge on the role of ctDNA and ptDNA in the management of PC-CRC. The literature search was conducted using a variety of academic databases, including PubMed, Scopus, and Google Scholar, with the aim of identifying relevant studies published from 2010 to 2025. The search terms included “liquid biopsy”, “circulating tumor DNA”, “peritoneal tumor DNA”, “colorectal cancer”, and “peritoneal carcinomatosis”. Studies were selected based on their relevance to the topic, focusing on those that explored diagnostic, prognostic, and therapeutic implications of ctDNA and ptDNA in PC-CRC.

For studies on localized disease, clinical trials that reported data on peritoneal recurrence were specifically considered. In the context of advanced disease, data were extracted from studies involving the general population, focusing on those that examined ctDNA or ptDNA in mCRC. While the review does not adhere to the PRISMA guidelines for systematic reviews, it offers a comprehensive synthesis of existing literature and highlights key findings, challenges, and future directions in this area of research.

Given the narrative approach, no strict inclusion or exclusion criteria were applied. The quality of the studies was assessed based on the relevance and reliability of the reported data, with particular attention given to studies providing insights into recurrence rates, treatment response, and molecular profiling through liquid biopsy techniques.

## 3. Circulating Tumor DNA: Definitions and Detection Assays

Cell-free DNA (cfDNA) represents most of the genomic material released in the bloodstream by normal cells and can be detected in plasma in concentrations that can vary from 1 to 10 ng/mL [33,34,35], with mean size around 180 base pair (bp) [36,37,38]. These fragments are released through active secretion or during cellular death by leukocytes, that are considered the main source of cfDNA in plasma [39,40].

In cancer patients, a small fraction within the total pool of cfDNA originates from tumor cells and is referred to as circulating tumor DNA. This is characterized by a different length (<145 bp) [41] and a very short half-life in bloodstream (15 min–2.5 h) [42,43], which makes it a good biomarker of real-time disease presence. As for cfDNA, the plasma concentrations of ctDNA show important variations, according to several features like tumor characteristics (e.g., primary tumor location and metastatic sites), tumor microenvironment (including micro-vascularization and immune infiltration), and patient clinical characteristics (e.g., surgical procedures, acute trauma, stroke, or myocardial infarction can increase the shedding of cfDNA and ctDNA) [44]. Pre-analytical and analytical features also have to be considered in the interpretation of ctDNA analysis, with the aim of guaranteeing reliability of results [45]. While these issues have been thoroughly reviewed elsewhere [46,47], an extensive explanation is beyond the purpose of this review. The fundamental pre-analytical and analytical conditions to consider in interpreting ctDNA results are briefly summarized in the next paragraphs.

### 3.1. Pre-Analytical Issues

Given the fragility of ctDNA fragments, it is important to establish a workflow that minimizes contamination and lysis. First, a large diameter needle (≤21 G) should be used to reduce the risk of damage. Second, collection tubes should be selected based on the time interval between sampling and plasma processing [48,49]. K_2_EDTA tubes are cost-effective and broadly available; they can be stored at room temperature for a maximum of 6 h or at 4 °C for maximum 24 h. In contrast, cell-stabilizing tubes, though more expensive, can be stored at room temperature for up 7 days [50].

The next essential step is plasma isolation through sequential centrifugation of the tubes at specific speed and temperatures [51]. In addition to plasma, the buffy coat (approximately 1%) should also be stored, as it can be used to assess germline mutations and the presence of mutations linked to clonal haematopoiesis of indeterminate potential (CHIP) [52,53], which can affect ctDNA analysis results [54].

### 3.2. Analytical Methods

The analytical methods for ctDNA analysis can be classified into two main categories: polymerase chain reaction (PCR)-based and next-generation sequencing (NGS)-based techniques [55].

The most used PCR-based method is digital PCR (dPCR), which splits the sample into micro-compartments, allowing each to act as an independent reaction site. This method offers exceptional sensitivity (<0.01%) and is particularly effective for detecting low-abundance ctDNA [56].

NGS-based techniques offer a broader range of applications, allowing whole-genome sequencing or targeted panels that cover specific hotspots or multiple genes. NGS detects a wide range of mutations, including deletions, rearrangements, and copy number alterations, with a limit of detection as low as <0.01 variant allele frequency (VAF) [57,58,59]. NGS’ main disadvantages include high costs, complex data analysis, and longer turnaround times, which can pose significant challenges in situations requiring rapid treatment decisions.

To increase sensitivity of ctDNA detection, especially for minimal residual disease (MRD), two strategies have been developed: “tumor-informed” and “tumor-agnostic” approaches [60]. In the “tumor-informed” approach, sequencing of the primary tumor tissue is first required to identify specific mutations specific to the tumor, which are then tracked in the plasma [61]. In contrast, the “tumor-agnostic” approach directly tests for a predefined panel of well-characterized mutations in plasma, without prior tumor sequencing [62]. The tumor-informed approach reduces the risk of false negatives but increases turnaround time, while the tumor-agnostic approach is faster and more cost-effective but may be less sensitive [60].

In recent decades, significant technological advances in the field of liquid biopsy have accelerated ctDNA research, leading to important achievements in demonstrating its clinical utility, particularly in CRC patients.

Liquid biopsy can potentially be applied at every stage of cancer management, from early detection to monitoring treatment responses. However, the choice of analytical method depends on the specific clinical question, as each technique has its own strengths and limitations regarding sensitivity, specificity, cost, and turnaround time [45].

Despite these advancements, peritoneal carcinomatosis has historically been identified as one of the most significant challenges in ctDNA analysis and result interpretation. In PC-CRC, the challenges in ctDNA detection are exacerbated by the low release of tumor DNA into the bloodstream due to the unique biology of peritoneal metastases. As a result, blood-based tests may fail to detect ctDNA, leading to false-negative results. Alternative sources of tumor DNA, such as peritoneal fluid, may offer more reliable markers for this patient population. In the following sections, we will explore the clinical applications of liquid biopsy in early and advanced CRC, with a specific focus on the available data for PM-CRC, aiming to evaluate the real impact of this scenario on ctDNA analysis outcomes.

## 4. ctDNA in Early-Stage CRC: Detection of Minimal Residual Disease

In early stage, liquid biopsy has proven to be a powerful tool for detecting “minimal residual disease” (MRD)—the presence of cancer-derived genomic material or circulating tumor cells (CTCs) when the primary tumor has been completely removed and is no longer detectable with standard imaging techniques [63].

Clinical studies involving CRC patients have significantly contributed to the field of ctDNA-based MRD detection and surveillance, possibly more than any other tumor type. Observational studies have evaluated the analytical validity (i.e., the accuracy and reliability of test measurement, including sensitivity, specificity, and reproducibility) and clinical validity (i.e., how consistently and accurately the test predicts outcomes such as recurrence or pathological response) [62,64,65,66,67]. These efforts have paved the way for randomized clinical trials to assess the clinical utility of ctDNA analysis—determining whether it can guide treatment decisions and improve patient care by reducing the use of adjuvant chemotherapy, lowering costs, and enhancing quality of life in colon cancer management during the adjuvant phase [60]. However, despite extensive evidence on ctDNA, MRD detection concerning peritoneal relapse remains underexplored. Here, we summarize the main findings available to date.

The phase II DYNAMIC-II trial is the landmark study that evaluated the clinical utility of ctDNA testing in early stage colon cancer. Overall, 455 stage II colon cancer patients were randomized post surgery (1:2 ratio) to either standard risk-factor-guided chemotherapy (i.e., tumor stage, tumor differentiation, node sampling < 12, lymphovascular invasion, tumor perforation, or bowel obstruction) or ctDNA-guided management. In the ctDNA-guided group, ctDNA-positive patients received adjuvant chemotherapy, while ctDNA-negative patients did not. The trial aimed to determine if ctDNA-guided management was non-inferior to standard management regarding 2-year recurrence-free survival (primary endpoint) and whether this approach could reduce chemotherapy use (secondary endpoint) [68]. The results confirmed both hypotheses. The updated 5-year analysis demonstrated a relapse-free survival rate of 88.3% with ctDNA-guided management versus 87.2% with standard management (HR, 1.01; 95% CI 0.56–1.81). The 5-year overall survival rates were 93.8% in the ctDNA-guided group and 93.3% in the standard group [69]. These findings suggest that ctDNA-guided decision-making can safely avoid adjuvant chemotherapy in nearly half of the patients who would otherwise receive it [70].

Nonetheless, the DYNAMIC-II trial highlighted some limitations of this approach, particularly the rate of false negative (FN) results. The CRC panel (Safe-Sequencing System) used in the trial had a sensitivity of 57.1%, specificity of 96.8%, positive predictive value (PPV) of 80%, and negative predictive value (NPV) of 90.9%. This translates to a significant rate of FN cases. Of note, the trial included double liquid biopsy analysis (extraction at week 4 and week 7) with the aim to reduce the rate of FN. To improve the accuracy of ctDNA to detect MRD, an exploratory analysis of the DYNAMIC-II trial utilized a whole exome sequencing panel with higher sensitivity (85.7%) and specificity (100%), yielding a PPV of 100% and an NPV of 96.9%. Consequently, the number of false negatives was significantly reduced [69].

In order to better understand the false negative rate, ctDNA results were matched to the type of recurrence (locoregional vs. distant metastases). Interestingly, while patients with distant metastases were ctDNA-negative in 53% of cases, all locoregional-only recurrences were consistently ctDNA-negative [71]. Despite low numbers, these data suggested that ctDNA was not predictive of locoregional recurrence. Although data on specific distant recurrence sites were unavailable for this trial, insights were gleaned from three other key clinical studies.

The PEGASUS trial evaluated a genetic plus epigenetic tumor-agnostic assay (Lunar 1.2; Guardant Health, Inc., Palo Alto, CA, USA) in patients with radically resected, high-risk stage II (*N* = 14) and stage III (*N* = 121) colon cancer. The study aimed to assess the feasibility of using liquid biopsy to guide adjuvant treatment based on MRD detection, mitigate false negatives through double liquid biopsy extraction and analysis, and explore treatment escalation for ctDNA-positive patients post-capecitabine-based chemotherapy. The primary endpoint was the false-negative rate [72]. At a median follow-up of 21 months, 22 relapses were observed: 10 were ctDNA-negative, and 12 were ctDNA-positive. Lung (*N* = 4) and peritoneal (*N* = 2) recurrences were associated with persistent postoperative ctDNA negativity in all cases [72].

The CIRCULATE-Japan GALAXY study provided additional insights into false negativity in the adjuvant setting. This observational study monitored ctDNA MRD status by using a tumor-based technology (Signatera, Austin, TX, USA) in patients with stage II–IV CRC. Interim analyses revealed a strong correlation between post-surgical ctDNA positivity and high recurrence risk compared to other clinicopathological features [73]. In a recent update with a median follow-up of 23 months (range 2–49 months) involving 2240 stage II–III resectable colon cancer or stage IV CRC patients, ctDNA positivity in the MRD window (2–10 weeks post-surgery) or surveillance period (4 weeks post adjuvant chemotherapy) was significantly associated with poor outcomes [74]. Overall, metastatic sites exhibited lower ctDNA positivity in the MRD window compared to the surveillance period. Peritoneum-only recurrence showed the most pronounced differences: ctDNA positivity rose from 42% in the MRD window to 84% during surveillance. Additionally, the authors introduced the concept of “transient clearance”, observed in three patients, one of whom had peritoneal recurrence. Unlike sustained ctDNA clearance, which correlated with favorable outcomes [24-month disease-free survival (DFS): 89.0%; OS: 100.0%], transient clearance was linked to poor prognosis (24-month DFS: 3.3%; OS: 82.3%). Clinical recurrences or death were eightfold higher in transient versus sustained clearance cases (86.21% vs. 10.29%), with one of two ctDNA conversions aligning with the end of adjuvant chemotherapy [74].

Henriksen et al. also observed delayed serum conversion of ctDNA in their nationwide Danish cohort study, which included 851 stage II–III CRC patients treated with curative intent [75]. For each patient, plasma samples were collected after radical surgery (up to 60 days after surgery) and, in a subgroup of patients (*N* = 246), every 3–4 months for up to 3 years. Digital PCR (BioRad, Hercules, CA, USA) was used for analysis. Notably, patients with peritoneal-only recurrence demonstrated a low ctDNA detection rate (20%) during the post-operative window. Although ctDNA positivity increased in the following months, a comparison of ctDNA levels between metastatic sites within three months of recurrence revealed a tendency toward lower median ctDNA levels in patients with peritoneal metastases compared to other metastatic sites, in line with findings from studies in advanced setting that will be discussed later in this review. However, due to the small sample size, no further statistical analysis could be performed.

Finally, Shah et al. presented the final analysis of the BESPOKE CRC study at the 2025 ASCO Gastrointestinal Cancers Symposium. This multicenter, prospective, observational study assessed the ability of a tumor-informed personalized ctDNA assay (Signatera) to guide adjuvant treatment decisions in stage II/III CRC patients [76]. Once again, post-operative ctDNA positivity demonstrated its prognostic value for disease-free survival (DFS) in both stage II (HR 11.23) and stage III patients (HR 8.33). Furthermore, in the surveillance setting, ctDNA testing showed high sensitivity in detecting recurrence across various sites. The authors concluded that this held true even for low ctDNA-shedding sites, such as the peritoneum and lungs. However, relapses in the peritoneum were associated with significantly lower sensitivity rates (79%) compared to high ctDNA-shedding sites like the liver (96%) or lymph nodes (89%) [77].

To conclude, available evidence suggests that peritoneal-only relapses may be more closely associated with delayed ctDNA positivity rather than true negativity (Table 1). This distinctive behavior likely reflects a molecular biology that differs significantly from other relapse sites. Unlike other metastatic sites, peritoneal metastases are often less vascularized, resulting in a lower amount of ctDNA being shed into the bloodstream. Additionally, the peritoneal cavity itself may act as a physical barrier, preventing ctDNA from entering the systemic circulation as efficiently as in more vascularized areas [78]. These biological factors contribute to the difficulty in detecting ctDNA during early peritoneal recurrence, leading to a higher risk of false-negative results in this context.

These findings highlight the importance of longitudinal MRD monitoring after treatment completion to promptly detect transitions from MRD-negative to MRD-positive status, improving clinical decision-making [60]. The optimal timing and frequency of ctDNA testing still require further validation. Based on current evidence, it is reasonable to suggest that ctDNA testing could be performed at regular intervals (e.g., every 3 to 6 months) following initial treatment. For patients at higher risk of recurrence, particularly those with high-risk features, more frequent testing may be considered to detect MRD as early as possible.

Although the exact frequency and timing of ctDNA monitoring remain to be standardized, this tool could complement conventional imaging techniques, providing an additional layer of sensitivity to detect relapses before they become visible on traditional scans. However, further research is needed to establish protocols for integrating ctDNA testing into routine clinical practice.

## 5. ctDNA in Advanced Disease

Before the clinical applications in early disease, ctDNA emerged as a versatile and minimally invasive biomarker for precision medicine in advanced CRC setting [79]. ctDNA is recommended by European and American academic guidelines to detect baseline resistant mutations to anti-EGFR inhibitors (i.e., *RAS* and *BRAF* mutations), in particular when tissue testing is not feasible or urgent therapeutic decisions are needed [3,80]. It also offers valuable insights into the evolution of resistant clones to guide anti-EGFR treatment rechallenge [81]. Moreover, ctDNA may be used to monitor molecular responses to therapy, anticipating radiological progression [82].

While the role of ctDNA has been extensively studied in mCRC with visceral metastases (mainly liver metastases), its application in PC-CRC remains less well-defined [31]. PC-CRC poses unique challenges [83], including a distinct tumor microenvironment [30] and lower ctDNA shedding [32], which could impact its detectability and clinical utility in this specific subgroup of patients. A deeper understanding of the biology of ctDNA in PC-CRC is essential for the implementation of ctDNA in this clinical scenario [84]. Table 2 summarizes the key findings from the studies discussed in the following sections.

### 5.1. Concordance Rates Between Tissue and Plasma Mutational Landscapes in PC-CRC

One of the key strengths of ctDNA is its high accuracy in capturing the tumor’s mutational landscape in real time [57]. Several studies have consistently shown a high degree of concordance between mutations detected in DNA extracted from tumor tissue and ctDNA isolated from peripheral blood from the same patient [44,85]. This strong evidence has led to the implementation of ctDNA for precision medicine in advanced disease. However, this correlation in tissue-plasma mutation detection is controversial in patients with peritoneal metastases [86].

Vidal et al. conducted a retrospective–prospective study to evaluate ctDNA as an alternative for determining baseline *RAS* status and monitoring emerging mutations during therapy in mCRC patients [87]. The study included 115 patients, of whom 27 (23.5%) were with peritoneal metastases. Both primary tumor tissue and baseline blood samples were analyzed using a dPCR-based method (OncoBEAM™ *RAS* CRC). The overall concordance between tissue and plasma *RAS* status was 93%. The overall concordance between tissue and plasma *RAS* status was 93%, suggesting that ctDNA can accurately reflect the mutations in tumors from other metastatic sites. However, discrepancies were observed in patients with peritoneal metastases. Specifically, two patients showed wild-type ctDNA despite having *RAS*-mutated tumor tissue (*KRAS* codon 12 mutations) and peritoneal metastases, while other patients with *RAS*-mutated ctDNA but wild-type tissue had metastases in the liver or lungs, but not the peritoneum. These results suggested that ctDNA detection might be less reliable in peritoneal carcinomatosis compared to other sites of metastasis, such as the liver and lungs [87].

**Table 2 cancers-17-01461-t002:** Observational studies on ctDNA or ptDNA detection in PC-CRC patients.

Reference	No. Patients	No. PC-CRC	Study Design	Analytical Method	Samples Analyzed	Key Findings
Vidal *Ann Oncol* 2017 [87]	115	27	Retrospective/ Prospectice Observational	dPCR (OncoBEAM™)	Baseline Tissue and Plasma	mVAF_PERITONEUM_ < mVAF_VISCERAL_
Hofste *Dis Colon Rectum* 2023 [88]	53	6	Retrospective Observational	NGS (Illumina)	Pre- and post-op plasma	mVAF_PERITONEUM_ < mVAF_VISCERAL_
Baumgartner *Ann Surg Oncol* 2018 [89]	80	11	Prospective Observational	NGS (Guardant Health)	Pre-op Plasma	High pre-op ctDNA levels associated with poor outcomes.
Kagawa *Clin Cancer Res* 2021 [90]	221	25	Retrospective Observational	dPCR (OncoBEAM™)	Baseline Tissue and Plasma	High concordance rate between tissue and plasma; mVAF_PERITONEUM_ < mVAF_VISCERAL_
Sullivan *Ann Surg Oncol* 2023 [91]	279	115	Retrospective Observational	NGS (Guardant Health)	Pre-treatment plasma	mVAF_PERITONEUM_ < mVAF_VISCERAL_
Baumgartner *Ann Surg Oncol* 2020 [92]	71	16	Prospective Observational	NGS (Guardant Health)	Pre- and post-op plasma	Pre- or Post-op high ctDNA levels correlate with poor outcomes
Dhiman *Ann Surg* 2023 [93]	33	13	Retrospective Observational	NGS (Signatera)	Post-op plasma	High post-op ctDNA levels associated with poor outcomes.
Beagan *J Clin Med* 2020 [94]	30	30	Retrospective Observational	ddPCR (BioRad)	Surgical tissue; Pre- and post-op plasma	Pre-op ctDNA does not correlate with PCI; Pre-op ctDNA associated with poor outcomes.
Loupakis *JCO PO* 2021 [95]	112	16	Retrospective Observational	NGS (Signatera)	Post-op plasma	Post-op ctDNA detection associated with poor outcomes
Lopez-Rojo *Ther Adv Med Oncol* 2020 [96]	26	8	Prospective Observational	ddPCR (BioRad)	Pre- and post-op plasma; Pre- and post-op peritoneal fluid	ctDNA detectable in all peritoneal fluid samples; Post-op ctDNA negativity associated with good outcomes
Van’t Eve *J Pathol Clin Res* 2021 [97]	120	20	Prospective Observational	ddPCR (BioRad)	Pre-op plasma; pre-op peritoneal fluid	ctDNA detectable in all peritoneal fluid samples. Peritoneal fluid mMAF_PERITONEUM_ > Peritoneal fluid mMAF_VISCERAL_

This table summarizes the key findings from the current literature on ctDNA in peripheral bloodstream or ptDNA in peritoneal fluid in PC-CRC patients. Despite growing interest in this specific context, all available evidence comes from either retrospective or prospective observational studies, which include various metastatic sites, with peritoneal carcinomatosis representing only a small proportion of the overall patient population. Abbreviations: ctDNA, circulating tumor DNA; dPCR, digital polymerase chain reaction; ddPCR, droplet dPCR; MTM, mean tumor molecules; NGS, next-generation sequencing; No, number; PC-CRC, peritoneal carcinomatosis from colorectal cancer; PCI, peritoneal carcinomatosis index; ptDNA, peritoneal tumor DNA.

A similar trend in ctDNA levels across metastatic sites was observed by Hofste et al. in a small retrospective study of oligo-metastatic CRC patients (*N* = 53) undergoing radical surgery [88]. Among them, six patients had peritoneal-only disease, which was associated with lower pre-operative ctDNA levels compared to those with liver metastases.

Baumgartner et al. examined the role of ctDNA in peritoneal metastases from multiple cancer types [89] by analyzing 80 patients with resectable peritoneal metastases, including 11 mCRC patients, who underwent CRS + HIPEC. This study reported a high overall concordance rate (96.7%) between tissue and plasma DNA. However, the positive concordance rate—where mutations were detected in both tissue and ctDNA—was significantly lower at 35.3%, compared to the negative concordance rate of 96.6%. This suggests that the agreement was mostly found in cases where mutations were absent, and that the positive concordance was less reliable in patients with peritoneal metastases.

Further evidence was provided by Kagawa et al., investigating the concordance of *RAS* status between liquid and tissue biopsies in 216 mCRC patients [90], including 25 with peritoneal-only metastases. In this subset of patients, the concordance rate was 94%, supporting the test’s clinical accuracy (dPRC, OncoBEAM™). Interestingly, higher concordance rates were observed in cases with peritoneal lesions ≥ 20 mm in diameter, suggesting that lesion size, and consequently tumor burden, may play a critical role in ctDNA—tissue concordance.

Lastly, Sullivan et al. analyzed through an NGS test (Guardant Health 360) 279 patients with gastrointestinal tumors and peritoneal metastases, including 115 mCRC [91]. ctDNA levels were 2.5 times lower in peritoneum-only cases compared to those with visceral metastases (*p* < 0.01), and tissue–plasma concordance was notably lower (18%) than previously reported through the same NGS test. However, methodological limitations caution against overinterpreting these results.

These findings collectively suggest that ctDNA exhibits good concordance with tumor tissue. In the particular case of peritoneal metastases, controversial results have been published. Results from Vidal [87], Baumgartner [89], and Sullivan [91] indicate that the presence of peritoneal metastases may negatively impact ctDNA—tissue concordance, resulting in lower positive concordance rates compared to patients with non-peritoneal metastases. Moreover, according to Kagawa et al. [90], the extent of peritoneal involvement also influences concordance, with larger lesions yielding better agreement between tissue and plasma than smaller ones. Importantly, the small number of PC-CRC patients in all studies limits the generalizability of the findings. 

In conclusion, while ctDNA may be a reliable tool for characterizing the mutational landscape of PC-CRC, further studies with larger and dedicated cohorts are needed, as well as research to better understand biological characteristics of ctDNA and PC-CRC.

### 5.2. The Prognostic Role of ctDNA in PC-CRC Patients

Another critical aspect of ctDNA is the prognostic implications of ctDNA baseline levels as well as ctDNA dynamic changes during treatment [46].

Baumgartner et al. explored the prognostic role of ctDNA in a cohort of peritoneal metastases from multiple cancer types [89]. Pre-operative ctDNA was detectable in 38.8% patients, with *TP53* (25.8%) and *KRAS* (11.3%) as the most frequently detected genes. Here, low pre-operative ctDNA levels (cut-off: 0.25% of the detected cfDNA) were associated with significantly longer progression-free survival (PFS) compared to high levels: 15.0 vs. 7.9 months (HR 3.23, 95% CI 1.43–7.28; *p* = 0.005), respectively.

The same research group analyzed peri-operative ctDNA dynamics in 71 patients undergoing surgery for peritoneal metastases. Of these, 16 patients had CRC as primary tumor [92]. Again, a targeted NGS-based approach was adopted (Guardant Health 360) and ctDNA detection rates were consistent with those reported in previous studies; indeed, ctDNA was detected in 39.4% of cases pre-operatively and in 52% post-operatively. Notable changes in the mutational landscape were observed following surgery. While the number of deleterious mutations remained relatively stable (20 pre-operative vs. 22 post-operative), the rate of variants of unknown significance increased post-operatively (15 to 22). This study further underscored the prognostic value of ctDNA levels: high ctDNA levels both pre- and post-surgery were associated with worse progression-free survival (PFS) compared to low levels of ctDNA. Median PFS was 4.8 months versus 19.3 months pre-operatively (*p* < 0.001) and 9.2 months versus 15.0 months post-operatively (*p* = 0.049).

A retrospective study by Dhiman et al. assessed the role of ctDNA in predicting recurrence in patients with peritoneal metastases from CRC (*N* = 13) or appendiceal adenocarcinoma (*N* = 20) undergoing curative-intent CRS + HIPEC [93]. Blood samples were collected 4–6 weeks postoperatively and subsequently every three months for one year and analyzed with an NGS-based assay (Signatera). Median ctDNA levels—evaluated as mean tumor molecules per milliliter (MTM/mL)—in patients with peritoneal recurrence (*N* = 13) were numerically lower than in patients with visceral recurrence (*N* = 3): 0.94 MTM/mL vs. 199.3 MTM/mL, respectively. Consistent with Baumgartner’s findings [92], high levels of ctDNA after radical peritoneal surgery were linked to worse outcomes (HR for DFS 3.67; 95% CI: 1.06–12.66; *p* = 0.03).

Beagan et al. provided further insights with a prospective study of 30 PC-CRC patients undergoing CRS + HIPEC [94]. Blood samples were collected pre-operatively and during follow-up, analyzed for bespoke mutations using targeted NGS and droplet digital PCR (ddPCR; BioRad). Pre-operative ctDNA positivity (33%) was equally observed in patients completing surgery and in patients undergoing open-close procedures due to extensive disease. Therefore, pre-operative ctDNA did not correlate with PCI, supporting the notion that ctDNA levels do not reflect peritoneal disease burden. Post-operative ctDNA variations were closely linked to recurrence patterns. Of all systemic relapses, four out of five patients had detectable post-operative ctDNA, while only one of eight with locoregional recurrence showed ctDNA positivity. None of the recurrence-free patients had detectable ctDNA during follow-up.

In the PREDATOR study, Loupakis et al. evaluated the prognostic value of post-operative ctDNA in 112 mCRC patients after resection of metastases [95]. Plasma samples were collected after surgery (median 27 days) and analyzed using an NGS-based approach (Signatera). Overall, 16 patients (14.3%) had peritoneal lesions; however, the type of surgery or whether different metastatic sites were involved at the same time has not been reported. Post-operative ctDNA positivity strongly predicted worse outcomes for DFS (HR 5.8, *p* < 0.001) and OS (HR 16; *p* < 0.001). At data cut-off, 96% of post-operative ctDNA-negative patients were alive, compared to 52.4% in post-operative ctDNA-positive subgroup.

In conclusion, a positive ctDNA status before and/or after CRS + HIPEC represents a strong negative prognostic factor, while complete ctDNA clearance following radical peritoneal treatments is associated with prolonged survival rates [92,95]. Furthermore, data suggest that ctDNA levels do not correlate with radiological disease burden in PC-CRC patients [93]. Indeed, ctDNA levels may be more influenced by the intrinsic biology of the metastatic site rather than the overall tumor load [87,94].

These findings suggest that PC-CRC might behave more like a localized disease rather than a systemic one. To advance on our understanding of PC-CRC biology, alternative sources of tumor-derived DNA should be explored as complementary or more powerful biomarkers. In this sense, peritoneal fluid emerged as a promising candidate, offering a more localized and potentially richer source of tumor DNA.

## 6. Peritoneal Tumor DNA: A Potential Biomarker in the Management of Peritoneal Carcinomatosis

The detection of tumor DNA in peritoneal fluid (ptDNA) may offer significant advantages, particularly due to the unique anatomical characteristics of the peritoneal cavity [98].

Peritoneal metastases are in direct contact with the peritoneal fluid, which may explain why the concentrations of tumor DNA are often higher in peritoneal fluid than in peripheral blood. Additionally, the peritoneal–plasma barrier—a physical separation between the peritoneum and blood—can limit the release of ctDNA from the peritoneal cavity into the bloodstream [99]. As a result, a high concentration of tumor DNA may be found in the peritoneal cavity.

This “tumor DNA sanctuary” offers several advantages. The relatively low amount of cell-free DNA in peritoneal fluid enhances the ability to detect mutations, even at low frequencies. Specifically, peritoneal fluid has a reduced “background noise” typically caused by DNA released from healthy tissues, which often interferes with the analysis of plasma samples [52]. Since ptDNA is isolated from other body compartments, it can more precisely reflect the mutational landscape of the peritoneal disease, offering valuable insights into the disease biology [84].

Despite these advantages, there are clear limitations. Collecting peritoneal fluid is an invasive procedure [100], unlike blood sampling, which is simple and can be repeated without significant discomfort. Moreover, not all patients with peritoneal metastases develop ascites [8]. In such cases, peritoneal fluid can only be collected during surgery, either through peritoneal lavage or by using drainage tubes during post-operative recovery, which may dilute the concentration of ptDNA (Figure 1).

Although ptDNA collection may present logistical challenges, this biomarker could play a crucial role in the future of monitoring PC-CRC. The main areas of utilization for this tool include the following:Treatment Response Monitoring: ptDNA can be used to monitor the efficacy of treatments, such as chemotherapy, targeted therapies, or immunotherapy. By providing real-time tracking of tumor molecular changes, ptDNA enables dynamic assessment of treatment response.Recurrence Prediction: ptDNA can be particularly useful for early detection of recurrence, even in patients who do not show clinical signs of progression. Since ptDNA directly reflects genetic alterations in the peritoneal microenvironment, it may be more sensitive than traditional imaging techniques (e.g., CT, MRI) in detecting residual disease. This sensitivity allows ptDNA to predict recurrences before they become clinically visible, enabling timely interventions.Personalized Treatment: another key advantage of ptDNA is its integration with ctDNA to provide a personalized approach to therapy. While ctDNA offers global insights into tumor dynamics, ptDNA provides more precise information on peritoneal disease, which is often the primary site of progression in PC-CRC. Combining these two biomarkers allows clinicians to better understand tumor burden and treatment response, leading to more informed therapeutic decisions, such as intensifying therapy for patients with developing recurrences or de-escalating treatment for those with a complete response.

The evidence currently available is limited. Nevertheless, recently published studies highlight the potential role of ptDNA in managing CRC patients with peritoneal metastases. Here, we report the key findings.

Lopez-Rojo et al. analyzed peritoneal fluid and blood from 11 patients with advanced CRC, pseudomyxoma peritonei, or localized CRC at high risk of peritoneal metastases undergoing CRS + HIPEC [96]. Blood was collected before and two days after surgery, while peritoneal fluid was collected during and after surgery. Using ddPCR (BioRad), they detected tumor DNA in 45.5% of blood samples and 54.5% of peritoneal fluid samples. Patients with detectable tumor DNA in either blood or peritoneal fluid after surgery had worse outcomes. Median DFS was 8.3 months, and mOS was 22.8 months in patients with undetectable tumor DNA (ctDNA or ptDNA). In comparison, patients with undetectable tumor DNA after surgery had much better outcomes, with mDFS of 35.4 months and a 3-year OS of 80%.

The second study was conducted by Van’t Erve et al. and involved 20 mCRC patients with peritoneal metastases only, within the CRC-PIPAC trial [97]. All patients underwent blood and peritoneal fluid collection before treatment, and ddPCR (BioRad) was used to detect *KRAS* and *BRAF* mutations. Mutations were detected in all peritoneal fluid samples, but only in 1 out of 5 blood samples. The concentration of tumor DNA (measured as MAF) was much higher in peritoneal fluid than in blood (16.4% vs. 0.28%; *p* < 0.0001). This confirms that peritoneal fluid may be a better source of tumor DNA for patients with peritoneal involvement. However, if ctDNA is detected in blood, it may indicate the presence of hidden systemic metastases in other sites and a higher risk of disease progression beyond peritoneal metastasis. Unfortunately, the study did not report correlations between ptDNA levels and survival outcomes.

Despite the small size, these proof-of-concept studies underscore the potential of ptDNA and ctDNA in guiding the treatment of peritoneal metastases from colorectal cancer. A combination of blood and peritoneal fluid analyses could stratify patients into different risk groups, helping clinicians to tailor treatments and to improve outcomes.

## 7. Future Perspectives in the Management of Peritoneal Carcinomatosis from Colorectal Cancer

While results are preliminary, both ctDNA and ptDNA are expected to play a key role in the future management of PC-CRC patients. Despite significant progresses in the treatment of these patients, clinicians still face challenges in accurately predicting which patients require further treatments, which patients may experience recurrence after radical surgery and HIPEC, or which patients can avoid unnecessary adjuvant systemic therapies [8].

During the diagnostic phase, ctDNA (and ptDNA when feasible) should be used to identify actionable biomarkers to guide treatment decisions, similar to the approach used for mCRC patients in the first-line setting [3]. These biomarkers could potentially also be used to evaluate tumor molecular response during and after systemic treatments [101].

Another potential clinical application of ctDNA and ptDNA is MRD detection after radical treatments like CRS + HIPEC [102]. As already noted, current evidence suggests that peritoneal metastases should be regarded more as a loco-regional rather than a systemic disease. Accordingly, the presence of ctDNA in the blood of PC-CRC patients may be a marker of microscopic metastases elsewhere. Based on this rationale, four possible potential scenarios may be hypothesized after completing CRS + HIPEC (Figure 2):Systemic Molecular Residual Disease (sMRD+; ctDNA positive, ptDNA negative): CRS + HIPEC successfully controlled peritoneal disease, but micrometastatic disease in the bloodstream may confer a risk of relapse in organs other than the peritoneum (i.e., liver metastases). Adjuvant systemic chemotherapy may be needed to eradicate micro-metastases and achieve complete disease control.Locoregional Molecular Residual Disease (lMRD+; ctDNA negative, ptDNA positive): CRS + HIPEC failed in controlling the peritoneal disease. This scenario may call for other localized interventions in the peritoneum, such as second-look surgery or alternative treatments like pressurized intraperitoneal aerosolized chemotherapy (PIPAC) [103].Dual Molecular Residual Disease (dMRD+; ctDNA positive, ptDNA positive): in this case, the presence of ctDNA and ptDNA indicates residual disease locally and micro-metastases systemically. A shift to systemic therapies could be necessary to address the minimal residual disease at both loco-regional and systemic levels.No Evidence of Molecular Residual Disease (MRD-; ctDNA negative, ptDNA negative): this is the ideal scenario where both ctDNA and ptDNA are undetectable, indicating complete disease clearance both locoregional and systemic. In this case, no further treatment may be needed, and the patient can be monitored through surveillance.

These four potential models highlight how ctDNA and ptDNA could be integrated into precision medicine to better manage PC-CRC patients. However, they remain speculative, as they have not yet been clinically validated through prospective studies or clinical trials to assess their applicability in practice.

Therefore, future studies should focus on evaluating the sensitivity and specificity of ctDNA and ptDNA in detecting MRD, as well as their ability to predict recurrences and guide treatment decisions. While promising, the integration of MRD detection into clinical practice requires rigorous validation through large-scale and multicenter studies. Additionally, the role of ctDNA and ptDNA in guiding the timing and intensity of treatment for patients with advanced PC-CRC must be further explored.

Furthermore, to fully realize the potential of ctDNA and ptDNA in clinical practice, a coordinated, multidisciplinary approach is essential. This should involve close collaboration between oncologists, molecular pathologists, and laboratory specialists to ensure proper sample collection, analysis, and interpretation of results.

Practical recommendations for this multidisciplinary approach include the development of standardized protocols for ptDNA collection, which can be challenging due to the invasive nature of the procedure. These protocols should clearly define the appropriate techniques for fluid collection (e.g., peritoneal lavage, drainage tubes), as well as the best practices for sample handling and storage to maintain DNA integrity [104].

Additionally, molecular tumor boards should play a crucial role in interpreting ctDNA and ptDNA data, ensuring these biomarkers are seamlessly integrated into personalized treatment plans. Regular meetings of the tumor board can offer clinical insights, prioritize therapeutic approaches based on molecular data, and provide a framework for determining the most effective treatment regimens [105].

The establishment of a clear workflow for both ctDNA and ptDNA testing, along with regular molecular profiling, will help facilitate more precise and timely treatment adjustments. This, in turn, will lead to better patient outcomes.

## 8. Conclusions

Liquid biopsy techniques, particularly ctDNA and ptDNA, hold great promise for revolutionizing the management of PC-CRC [96,97]. These methods offer a non-invasive and dynamic approach to monitoring disease progression [87,89,90,91], detecting early recurrences [92,93], and assessing therapeutic responses [92,93,94,95]. While ctDNA has already demonstrated value in monitoring metastatic disease, ptDNA may provide more accurate and reliable results in PC-CRC, especially given the challenges of detecting ctDNA in peripheral blood [60].

Despite their potential, both ctDNA and ptDNA in PC-CRC are still in the early stages of clinical application and face several limitations. These include issues related to sensitivity, sample collection, and the standardization of methods. Furthermore, there is significant heterogeneity in methodologies used across studies, and many of the findings are based on small-scale studies. To address these challenges, large-scale clinical trials, prospective studies, and multicenter research are needed to validate these biomarkers and establish their reliable role in clinical practice [106].

Looking ahead, future research should focus on several key priorities: comparing ctDNA and ptDNA in PC-CRC, developing PC-CRC-specific assays, and exploring the utility of these biomarkers in guiding treatment decisions and monitoring MRD. Additionally, the integration of ctDNA and ptDNA into routine clinical practice will require a coordinated, multidisciplinary approach [104]. This includes close collaboration among oncologists, molecular pathologists, laboratory specialists, and clinical researchers to ensure proper sample collection, data interpretation, and therapeutic decision-making. Standardized protocols for sample collection and molecular profiling must be developed to ensure accurate, reliable results and to maximize the clinical benefit of these biomarkers.

In conclusion, while significant challenges remain, the future of ctDNA and ptDNA in managing PC-CRC patients is promising. Further research will be critical in translating these tools from the research setting to routine clinical practice, ultimately leading to more personalized and effective treatments for patients.

## Figures and Tables

**Figure 1 cancers-17-01461-f001:**
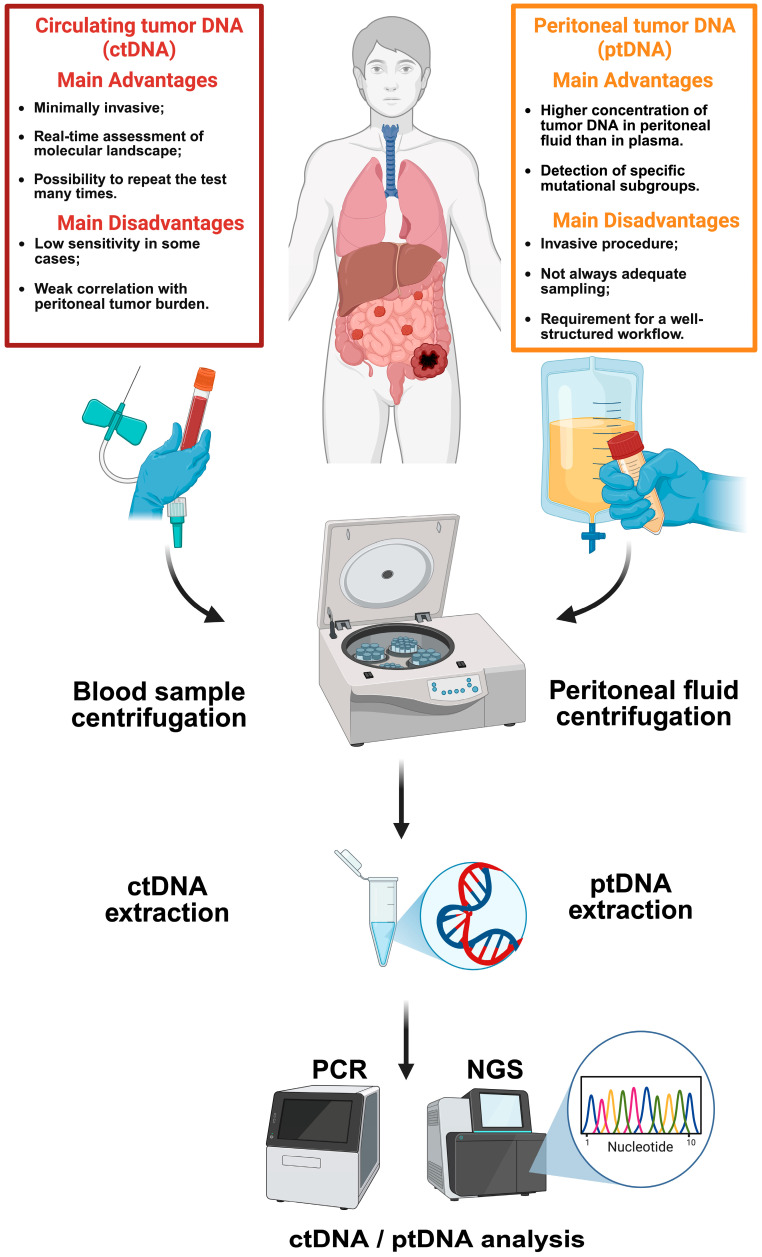
Key techniques for assessing tumor genomic profiles in peritoneal carcinomatosis from colorectal cancer patients and their main advantages and disadvantages. The figure summarizes the main advantages and disadvantages of two methods used to analyze tumor DNA: circulating tumor DNA (ctDNA) in blood and peritoneal tumor DNA (ptDNA) in peritoneal fluid. ctDNA, obtained from a simple blood draw, offers a less invasive approach but its sensitivity can be limited by the amount of tumor DNA present in the blood. ptDNA allows a more direct analysis of the peritoneal tumor but requires an invasive procedure and may be limited by the absence of peritoneal fluid. Abbreviations: ctDNA, circulating tumor DNA; MAF, mutant allele frequency; NGS, next-generation sequencing; PCR, polymerase chain reaction; ptDNA, peritoneal tumor DNA.

**Figure 2 cancers-17-01461-f002:**
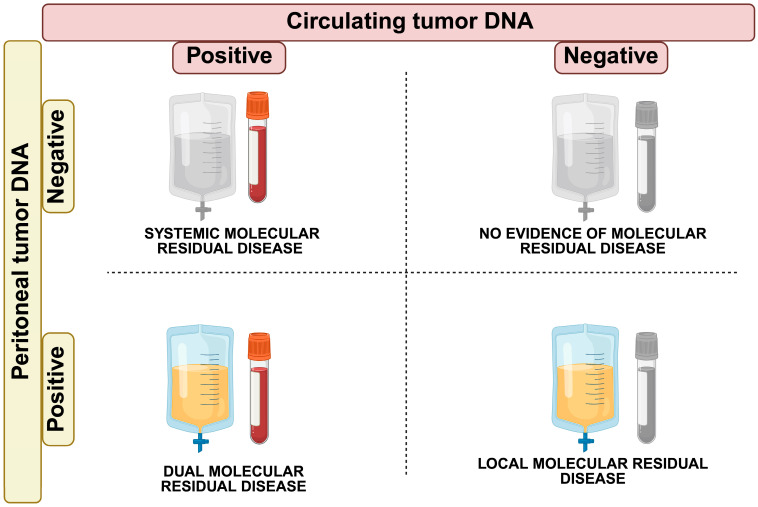
Post-operative Assessment of ctDNA in peripheral bloodstream and ptDNA in peritoneal fluid to guide the following treatments. The integration of ctDNA and ptDNA analyses after radical treatments such as cytoreductive surgery and hyperthermic intraperitoneal chemotherapy may guide personalized treatment decisions in patients with peritoneal carcinomatosis from colorectal cancer. By assessing molecular residual disease status both systemically and locally, clinicians could potentially tailor treatment strategies for individual patients. The blood vials represent liquid biopsy samples for ctDNA (gray if negative, colored if positive), and the drainage bags represent liquid biopsy samples for ptDNA (gray if negative, colored if positive).

**Table 1 cancers-17-01461-t001:** ctDNA-based studies reporting data on peritoneal recurrence.

Reference	Stage	Median Follow-Up (Months)	Countries	Study Design	Analytical Method	No. Patients	No. Peritoneal Relapses	Key Findings on Peritoneal-Only Recurrence
Lonardi *ESMO* *Congress* 2023 [72]	II (high risk)–III	21.7	Spain, Italy	Non-randomized interventional	NGS (Guardant Health)	135	2	Peritoneal metastases associated with persistent ctDNA negativity; Peritoneal metastases associated with pT4N1 stage.
Nakamura *Nat Med* 2024 [74]	II–III–IV	23	Japan	Prospective observational	NGS (Guardant Health)	2.240	115	Higher ctDNA positivity rate in the surveillance window compared to the MRD window for peritoneal metastases; Peritoneal metastases may be associated with “transient clearance”.
Henriksen *Ann Oncol* 2024 [75]	II–III	26	Denmark	Prospective observational	dPCR (BioRad)	851	10	Lower ctDNA levels observed in peritoneal metastases compared to visceral metastases.
Shah *ASCO Gastrointestinal Cancers Symposium* 2025 [77]	II–III	23.2	USA	Prospective observational	NGS (Signatera)	1.166	20	Lower sensitivity rates in peritoneal relapses (79%) compared to liver (96%) or nodes (89%).

The table summarizes key studies investigating the role of ctDNA in the context of peritoneal-only metastases in colorectal cancer. The studies span different stages of disease (II to IV) and employ various study designs and analytical methods to explore the behavior of ctDNA in patients with peritoneal metastases. Key findings highlight unique characteristics of ctDNA in this metastatic site compared to visceral metastases. Abbreviations: ctDNA, circulating tumor DNA; MRD, minimal residual disease; No, number; NGS, next-generation sequencing; dPCR, digital polymerase chain reaction.

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
