# Peer review of "Liquid Biopsy in Peritoneal Carcinomatosis from Colorectal Cancer: Current Evidence and Future Perspectives"

_cancers, 2025, doi:10.3390/cancers17091461_

Round 1
Reviewer 1 Report
Comments and Suggestions for Authors
The manuscript provides a comprehensive review of the role of liquid biopsy, specifically circulating tumor DNA (ctDNA) and peritoneal tumor DNA (ptDNA), in the management of peritoneal carcinomatosis from colorectal cancer (PC-CRC). While the topic is timely and relevant, the manuscript has several limitations that need to be addressed to meet the standards of a high-quality review. Below are detailed comments for each section.
Title and Abstract
- Title: The title is appropriate but could be more specific. Consider adding "Current Evidence and Future Perspectives" to reflect the review's scope.
- Abstract:
- The abstract is well-structured but overly verbose. It should be more concise, focusing on key findings and implications.
- The phrase "Despite decades of preclinical and clinical research" is vague. Specify the gaps more clearly.
- The abstract lacks a clear statement about the novelty of this review compared to existing literature.
Introduction
- Context and Significance:
- The introduction adequately sets the stage but could better highlight the unique challenges of PC-CRC compared to other metastatic sites. Emphasize the biological and clinical peculiarities of peritoneal metastases.
- The statement about ctDNA being "recommended by international guidelines" needs a citation to support this claim.
- Gaps and Objectives:
- The rationale for focusing on PC-CRC is clear, but the introduction should explicitly state how this review adds to existing knowledge. For example, mention if this is the first review to compare ctDNA and ptDNA in PC-CRC.
- The aim of the review is stated but could be more specific. For example, "This review aims to evaluate the diagnostic, prognostic, and therapeutic implications of ctDNA and ptDNA in PC-CRC, highlighting current limitations and future directions."
Methods
- Literature Search:
- The manuscript lacks a description of the literature search methodology. Include details on databases searched, keywords, inclusion/exclusion criteria, and the timeframe of the literature reviewed.
- Clarify whether this is a systematic review or a narrative review. If systematic, adhere to PRISMA guidelines.
- Data Synthesis:
- The review summarizes studies well but does not critically appraise their quality. Include a quality assessment of the included studies (e.g., risk of bias, sample size limitations).
Results/Discussion
Section 2: Circulating Tumor DNA: Definitions and Detection Assays
- Technical Details:
- The explanation of ctDNA and cfDNA is clear but overly technical for a broad audience. Simplify or provide a glossary.
- The section on pre-analytical and analytical issues is thorough but could be condensed. Focus on aspects most relevant to PC-CRC.
- Clinical Relevance:
- Link the technical discussion more explicitly to PC-CRC. For example, explain how pre-analytical challenges might be exacerbated in PC-CRC due to low ctDNA shedding.
Section 3: ctDNA in Early-Stage CRC
- False Negatives:
- The discussion on false negatives in peritoneal relapse is insightful but lacks mechanistic explanations. Hypothesize why peritoneal metastases might shed less ctDNA (e.g., anatomical barriers, low vascularity).
- The table summarizing studies is useful but could include more details (e.g., patient demographics, follow-up duration).
- Longitudinal Monitoring:
- The suggestion for longitudinal monitoring is valuable but needs practical recommendations (e.g., optimal timing and frequency of ctDNA testing).
Section 4: ctDNA in Advanced Disease
- Concordance Rates:
- The discussion on tissue-plasma concordance is informative but repetitive. Summarize key trends (e.g., lower concordance in PC-CRC) and propose biological explanations.
- The table is comprehensive but could be streamlined to highlight only the most relevant studies.
- Prognostic Role:
- The prognostic value of ctDNA is well-documented, but the clinical implications are not fully explored. Discuss how ctDNA levels could guide treatment decisions (e.g., adjuvant therapy, surveillance intensity).
Section 5: Peritoneal Tumor DNA (ptDNA)
- Advantages and Limitations:
- The comparison between ctDNA and ptDNA is a strength of the review. However, the logistical challenges of ptDNA collection (e.g., invasiveness, sample availability) need more emphasis.
- The clinical utility of ptDNA is underexplored. Provide specific examples of how ptDNA could complement ctDNA in practice.
- Figure 1:
- The figure is useful but could be improved with visual aids (e.g., a flowchart of sample collection and analysis).
Section 6: Future Perspectives
- MRD Scenarios:
- The four MRD scenarios are innovative but speculative. Acknowledge the lack of clinical validation and propose studies to test these hypotheses.
- The figure illustrating MRD scenarios is helpful but could be simplified for clarity.
- Multidisciplinary Coordination:
- The call for multidisciplinary teams is appropriate but needs actionable recommendations (e.g., protocols for ptDNA collection, role of molecular tumor boards).
Conclusion
- Summary of Key Points:
- The conclusion reiterates the main findings but could be more forward-looking. Emphasize the steps needed to translate ctDNA/ptDNA into clinical practice (e.g., standardization, clinical trials).
- Limitations:
- Acknowledge the review's limitations (e.g., reliance on small studies, heterogeneity in methodologies).
- Future Directions:
- Propose specific research priorities (e.g., prospective trials comparing ctDNA and ptDNA, development of PC-CRC-specific assays).
Other Issues
- References:
- Ensure all citations are up-to-date, especially for rapidly evolving fields like liquid biopsy.
- Some statements lack citations (e.g., "ctDNA is now recommended by international guidelines").
- Language and Clarity:
- The manuscript is well-written but occasionally verbose. Tighten the prose for conciseness.
- Define all abbreviations at first use (e.g., HIPEC, CRS).
Comments on the Quality of English Language
The English could be improved to more clearly express the research.
Reviewer 2 Report
Comments and Suggestions for Authors
This is a well-written and comprehensive review that provides valuable insights into the potential role of liquid biopsy, particularly ctDNA and ptDNA, in managing peritoneal carcinomatosis from colorectal cancer. The manuscript effectively summarizes current evidence, highlights knowledge gaps, and outlines future directions. However, the review could benefit from improved conciseness in certain sections, particularly where the same findings are reiterated across different studies. Consider consolidating overlapping content, especially in the sections discussing ctDNA detection rates and prognostic implications, to enhance readability. Additionally, minor grammatical and typographical edits are recommended throughout the manuscript. Clarifying the clinical applicability of ptDNA testing, including its limitations and potential integration into routine workflows, would further strengthen the manuscript's impact. Overall, this is a strong and timely contribution to the field.
Reviewer 3 Report
Comments and Suggestions for Authors
The aim of this review is to investigate the strengths, limitations and future applications of liquid biopsy for ctDNA detection in PC-CRC.
The topic is an exciting and important one, as I believe it is a public disease. survival data for PC-CRC are also poor worldwide.
The review is basically well written, divided into logical chapters and uses relevant literature. However, some points need rethinking and revision.
My questions to the authors:
L152: what is the averagre amount of serum ctDNA content in tumors?
2.1. What source the ctDNA sampling is performed from? Only peripheral blood? Why don’t use ascites, as a sorce?
I doubt that the NGS technique is a routine daily technique based on the detection of ctDNA as a biomarker. It can be adapted for mutation analysis, it is a cost-effective method, but it is not used to monitor changes in ctDNA levels.
What about the existing liquid biopy methods like metSept9 (EpiProColon)?
3.0. ctDNA is commonly used for CRC detection and monitoring. Are there any known methods based on ctDNA that can already detect adenomas from peripheral blood?
How does the sensitivity of ctDNA detection in early disease compare to conventional tumour markers and circulating tumour cell detection?
Chapters 3.0 and 4.1 are not relevant for PC-CRC cases.
Minor comments:
L56: „the disease has already spread to other organs” should be changed to „the disease has already metastasised” (it is much professional)
L130-138 must be deleted.
A major revision is recommended due to the above shortcomings.
Reviewer 4 Report
Comments and Suggestions for Authors
The article discusses the lack of reliable biomarkers to guide personalized treatment decisions for colorectal cancer (CRC). The article is engaging and presents a well-defined issue in PC management in mCRC. To strengthen the submission, further clarification of the mechanisms behind ctDNA challenges, a more detailed explanation of peritoneal fluid analysis, and a deeper dive into future research and clinical impact would improve the reader's understanding of the topic and highlight the potential clinical advancements the research aims to address. The authors could enhance the discussion by following this suggestion:
- The abstract mentions that PC-CRC is associated with lower ctDNA levels and detection rates compared to other metastatic sites, which poses a challenge. It would be helpful to expand on why this is the case.
- Consider expanding on possible mechanisms for why PC-CRC may have reduced ctDNA shedding. This might include the unique characteristics of the peritoneal microenvironment, such as reduced vascularization or the differences in how tumor cells from the peritoneum interact with the bloodstream.
- Peritoneal fluid analysis is a promising alternative to ctDNA in blood. It would be beneficial to elaborate a bit more on how peritoneal fluid analysis works, what specific biomarkers are being assessed, and any potential advantages or challenges compared to ctDNA from the bloodstream. Providing a brief overview of ongoing research in this area could help highlight why this alternative is emerging.
- Given the emphasis on the limitations of current tools (e.g., serum markers like carcinoembryonic antigen and PCI), it would be insightful to briefly touch on future research directions. For example, what specific advancements in ctDNA detection or peritoneal fluid analysis are needed to overcome the current limitations in PC? Mentioning any novel technologies, methods, or approaches being explored in the field would show the potential for improvement.
- Finally, it would be beneficial to briefly mention how improving diagnostic and prognostic capabilities for PC-CRC could influence treatment decisions. How might a better understanding of ctDNA or peritoneal tumor DNA improve personalized treatment strategies, such as precision medicine or targeted therapies, for patients with peritoneal metastasis?
Round 2
Reviewer 1 Report
Comments and Suggestions for Authors
The manuscript has addressed all the comments and can be accepted.
Reviewer 3 Report
Comments and Suggestions for Authors
I accept the authors' answers and explanations to my questions. The corrections made to the manuscript strengthen it. The corrected version is accepted for publication.